# Effects of microcompartmentation on flux distribution and metabolic pools in *Chlamydomonas reinhardtii* chloroplasts

**Anika Küken[1,2], Frederik Sommer[1†], Liliya Yaneva-Roder[1], Luke CM Mackinder[3‡], Melanie Höhne[1], Stefan Geimer[4], Martin C Jonikas[3§], Michael Schroda[1†], Mark Stitt[1], Zoran Nikoloski[1,2], Tabea Mettler-Altmann[1,5,6]***

[1]Max Planck Institute of Molecular Plant Physiology, Potsdam-Golm, Germany; [2]Bioinformatics Group, Institute of Biochemistry and Biology, University of Potsdam, Potsdam, Germany; [3]Department of Plant Biology, Carnegie Institution for Science, Stanford, United States; [4]Institute of Cell Biology, University of Bayreuth, Bayreuth, Germany; [5]Cluster of Excellence on Plant Sciences, Heinrich-Heine University, Düsseldorf, Germany; [6]Institute of Plant Biochemistry, Heinrich-Heine University, Düsseldorf, Germany

**\*For correspondence:**
tabea.mettler@hhu.de

**Present address:** [†]Molecular Biotechnology and Systems Biology, University of Kaiserslautern, Kaiserslautern, Germany; [‡]Department of Biology, University of York, York, United Kingdom; [§]Department of Molecular Biology, Princeton University, Princeton, United States

**Competing interests:** The authors declare that no competing interests exist.

**Abstract** Cells and organelles are not homogeneous but include microcompartments that alter the spatiotemporal characteristics of cellular processes. The effects of microcompartmentation on metabolic pathways are however difficult to study experimentally. The pyrenoid is a microcompartment that is essential for a carbon concentrating mechanism (CCM) that improves the photosynthetic performance of eukaryotic algae. Using *Chlamydomonas reinhardtii*, we obtained experimental data on photosynthesis, metabolites, and proteins in CCM-induced and CCM-suppressed cells. We then employed a computational strategy to estimate how fluxes through the Calvin-Benson cycle are compartmented between the pyrenoid and the stroma. Our model predicts that ribulose-1,5-bisphosphate (RuBP), the substrate of Rubisco, and 3-phosphoglycerate (3PGA), its product, diffuse in and out of the pyrenoid, respectively, with higher fluxes in CCM-induced cells. It also indicates that there is no major diffusional barrier to metabolic flux between the pyrenoid and stroma. Our computational approach represents a stepping stone to understanding microcompartmentalized CCM in other organisms.

## Introduction

Compartments of eukaryotic cells are surrounded by a single- or multiple-layer lipid membrane. Both eukaryotic and prokaryotic cells also include microcompartments that are not separated from the rest of the cell by a lipid membrane (for reviews, see *Giordano et al., 2005*; *Hyman et al., 2014*). In bacteria they are surrounded by protein shells (for reviews, see *Kerfeld and Melnicki, 2016*; *Yeates et al., 2010*). Such microcompartments may partition metabolic pools and enzymes; therefore, they can directly affect the operation of metabolic pathways. Microcompartments may serve diverse roles, from storage of special compounds (*Bazylinski and Frankel, 2004*), degradation of small molecules (*Bobik et al., 1999*), facilitation of enzyme clustering (*Castellana et al., 2014*), to regulating the activity of particular metabolic pathways.

The first described bacterial microcompartment was the carbon-fixing carboxysome in cyanobacteria (*Drews and Niklowitz, 1956*). It enables the cell to accumulate carbon dioxide ($CO_2$) in the vicinity of Rubisco, which enhances the carboxylation rate. The carboxysome is an essential part of the cyanobacterial carbon concentrating mechanism (CCM). There are two types of carboxysomes, the alpha and beta carboxysome, and the structure and function of both types have been well-

studied (*Kerfeld and Melnicki, 2016*). Loss of structural proteins results in carboxysome-less mutants that are unable to grow under ambient $CO_2$ conditions (*Berry et al., 2005*; *Ogawa et al., 1994*; *Woodger et al., 2005*). Two other microcompartmentalized pathways, namely, propanediol and ethanolamine utilization, have also been experimentally explored in bacteria (*Chen et al., 1994*; *Stojiljkovic et al., 1995*). The role of the propanediol utilization (Pdu) microcompartment in *Salmonella enterica* is the degradation of 1,2-propanediol, a product of anaerobic sugar breakdown, without the release of the degradation intermediate propionaldehyde. Propionaldehyde is toxic and, once in the cytosol, causes damage to DNA (*Sampson and Bobik, 2008*). A similar role was suggested for the ethanolamine utilization (Eut) microcompartment in the detoxification of acetaldehyde produced during ethonalamine catabolism (*Moore and Escalante-Semerena, 2016*). Microcompartments are also known in eukaryotes, including: metabolic compartments in liver (*Fujiwara and Itoh, 2014*) and muscle cells (*Saks et al., 2008*), and the pyrenoid in chloroplasts of green algae (*Gibbs, 1962*).

Despite these discoveries, it remains challenging to determine the implications of microcompartments for cellular physiology, and to study the function of microcompartments under different conditions that may induce or suppress their formation. This task is experimentally tedious and often not feasible due to challenges in separating the microcompartments (*Saks et al., 2008*). Here we present a combined experimental and mathematical approach to quantify metabolic exchange fluxes at the boundary of the pyrenoid in the chloroplast of the green alga *Chlamydomonas reinhardtii* under two environmental conditions, atmospheric $CO_2$ with an active CCM; and high $CO_2$, where the CCM is inactive.

Different CCMs have evolved in higher plants, algae and cyanobacteria to cope with the relatively low amounts of $CO_2$ in the atmosphere (currently 0.03–0.04%) and to compensate for the low affinity of Rubisco for $CO_2$ under these conditions (*Delgado et al., 1995*; *Tcherkez et al., 2006*). As already mentioned, CCM in cyanobacteria requires microcompartments called carboxysomes. In eukaryotic green algae, a microcompartment called the pyrenoid is crucial for the establishment of a CCM (*Caspari et al., 2017*; *Genkov et al., 2010*; *Figure 1*). There is no membrane or protein shell surrounding the pyrenoid which, like many of these non-membrane microcompartments (for review, see *Hyman et al., 2014*), was recently described as a liquid-like organelle formed by phase separation from the chloroplast stroma (*Freeman Rosenzweig et al., 2017*).

Pyrenoids are known to contain Rubisco (*Kuchitsu et al., 1988b*; *Kuchitsu et al., 1991*; *McKay et al., 1991*), Rubisco activase (*McKay et al., 1991*) and EPYC1, which has been proposed to be a structural protein in the pyrenoid (*Mackinder et al., 2016*). Immunolocalisation studies showed that about 40% and 90% of the total Rubisco is located in the pyrenoid under high $CO_2$ and atmospheric $CO_2$, respectively, while the rest is distributed in the stroma (*Borkhsenious et al., 1998*). Under ambient $CO_2$ when CCM is induced, experimental evidence suggests that the inorganic carbon ($C_i$, the sum of $CO_2$, $CO_3^{2-}$, $HCO_3$ and $H_2CO_3$) is actively transported across the membrane of the cell and accumulated as $HCO_3^-$ in the thylakoid lumen (for reviews, see *Grossman et al., 2007*; *Jungnick et al., 2014*; *Moroney and Ynalvez, 2007*; *Spalding, 2008*; *Wang et al., 2015*). These luminal regions have highly branched tubules that reach into the pyrenoid and may facilitate movement of bicarbonate, $CO_2$ and other small molecules (*Engel et al., 2015*). A carbonic anhydrase (CAH3), essential for the CCM, is localized to the lumen regions that pass through the pyrenoid and is thought to catalyze the dehydration of $HCO_3^-$ to $CO_2$, as the preferred form of $C_i$ at the low pH of the lumen (*Duanmu et al., 2009*; *Sinetova et al., 2012*). The resulting $CO_2$ may then diffuse back across the lumen membrane where it serves as concentrated substrate for the nearby Rubisco. A starch sheath, of unclear function, surrounds the pyrenoid, but only under ambient $CO_2$ conditions (*Kuchitsu et al., 1988a*; *Ramazanov et al., 1994*). The starch sheath was suggested to serve as a diffusional barrier for $CO_2$ and therefore potentially also larger molecules, such as metabolites, although the existing experimental evidence is inconsistent with this hypothesis (*Badger and Price, 1994*; *Villarejo et al., 1996*).

What does the location of Rubisco in the pyrenoid mean for the rest of the carbon fixation pathway? The current model of CCM assumes that, apart from Rubisco, the remaining enzymes of the Calvin-Benson cycle (CBC) are situated in the stroma (*Jungnick et al., 2014*). This implies that the substrate and product of the carboxylation reaction catalyzed by Rubisco, ribulose-1,5-bisphosphate (RuBP) and 3-phosphoglyccerate (3PGA), need to move in and out of the pyrenoid, respectively. This assumption is supported by immunolocalisation studies that failed to detect glyceraldehyde 3-

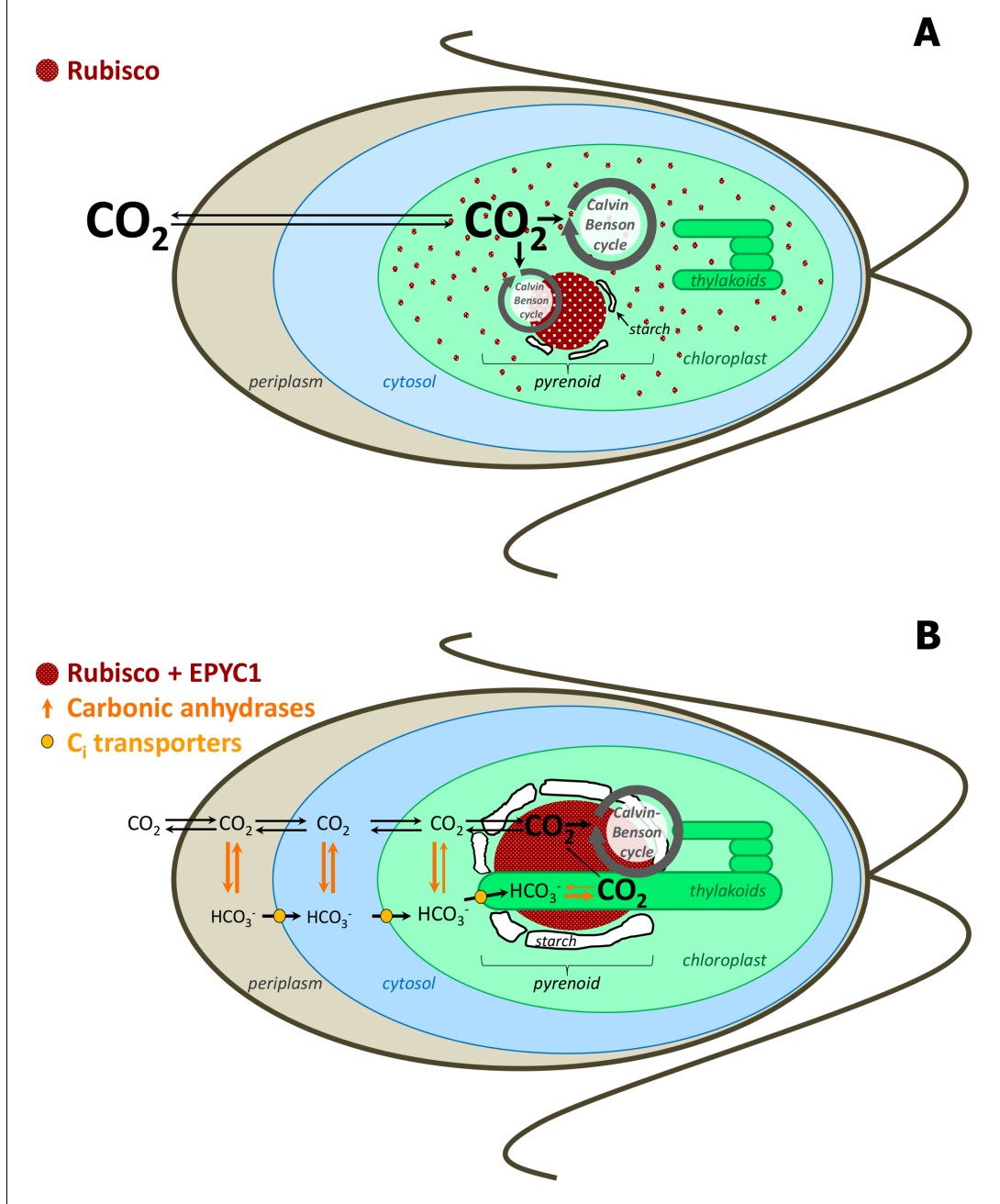

**Figure 1.** Simplified scheme of CBC cycle with and without carbon-concentrating mechanism (CCM) in *Chlamydomonas reinhardtii.* Under high $CO_2$ conditions, no CCM is established (**A**). After exposure to ambient $CO_2$, CCM is induced (**B**). As most of Rubisco and the other CBC enzymes are in the stroma under high $CO_2$, most CBC flux is in the stroma (big grey circle in the stroma) and only partly involves the pyrenoid (smaller grey circle) (**A**). As most of Rubisco is inside the pyrenoid under ambient $CO_2$, the CBC requires movement of selected metabolites between the stroma and pyrenoid (big grey circle) (**B**). To find out the exact differences of flux distribution between stroma and pyrenoid under these two conditions and how metabolites are exchanged between the two microcompartments were the aims of this study. Scheme adapted and simplified from *Borkhsenious et al. (1998)*, *Moroney et al. (2011)*, *Wang et al. (2011)*, *Engel et al. (2015)* and *Mackinder et al. (2016)*.

phosphate dehydrogenase (GAPDH), aldolase (FBA +SBA) catalyzing the two aldol reactions of the CBC that produce fructose-1,6-bisphosphate (FBP) and sedoheptulose-1,7-bisphosphate (SBP), ribose-phosphate isomerase (RPI) or phosphoribulokinase (PRK) in pyrenoids (*Kuchitsu et al., 1991*; *Suss et al., 1995*). Such methods, however, are not well suited to demonstrate absence, and experimental evidence is still lacking for the localization of the remaining CBC enzymes

(phosphoglycerokinase (PGK), triosephosphate isomerase (TPI), transketolase (TRK), fructose-1,6-bisphosphatase (FBPase), sedoheptulose-1,7-bisphosphatase (SBPase), and ribulose-phosphate epimerase (RPE)). Furthermore, the evidence for absence of CBC enzymes from the pyrenoid is not fully consistent as biochemical studies indicate PRK may be in close association with the pyrenoid (*Holdsworth, 1971*; *McKay and Gibbs, 1991*). Additionally, Rubisco is differently distributed under ambient and high $CO_2$ (*Borkhsenious et al., 1998*) implying that the flux of the CBC that takes place in the pyrenoid may differ between these two conditions.

Our study aims to experimentally localize all CBC enzymes, and to measure Calvin-Benson cycle intermediate levels under high and ambient $CO_2$. These data are then used in combination with mathematical modeling to estimate fluxes through the CBC in the stroma and the pyrenoid under these two $CO_2$ conditions. Our approach allows us to determine the exchange of fluxes at the boundary of the pyrenoid and to investigate the mode of transport of the exchanged metabolites.

## Results

### Distribution of Calvin-Benson cycle enzymes

*C. reinhardtii* CC1690 cells were grown under low $CO_2$ (LC), which fully induced the CCM (*Figure 2—figure supplement 1*). In addition, we obtained data from cells grown under high $CO_2$ (HC), where the induction of CCM was suppressed.

*C. reinhardtii* was fractionated to provide samples enriched for stroma proteins and for pyrenoid-associated proteins according to *Mackinder et al. (2016)*, followed by quantification of the abundance of enzymes involved in the CBC and starch synthesis, using either an enzymatic assay or shotgun proteomics (*Figure 2*, *Supplementary file 1A,B*). More than 61.8% of the Rubisco was found in the pyrenoid in LC grown cells, and about 21.8% in HC grown cells. Apart from GAPDH (8% in LC and 11% in HC grown cells) and PRK (13% in HC grown cells but <2% in LC grown cells) less than 2% of the other CBC proteins were detected in the pyrenoid-enriched fractions. The <2% of CBC proteins found in the pyrenoid-enriched fractions may represent experimental error, and resembled the distribution of phosphoglycerate mutase (PGM) and ADP-glucose pyrophosphorylase (AGPase) (0.6–1.9% in the pyrenoid *Supplementary file 1B*).

The localization of most of the CBC enzymes was further confirmed by confocal microscopy of proteins tagged with the yellow fluorescence protein Venus of cells grown under ambient $CO_2$ (*Nagai et al., 2002*; *Figure 3*, *Supplementary file 2*). This was of particular interest as different isoforms exists for some of the CBC proteins (Rubisco small subunit, GAPDH, FBPase and aldolase). The proteins encoded by the two Rubisco small subunit genes (*RBCS1* and *RBCS2*) both showed strong localization to the pyrenoid (discussed in more details below). PGK1, GAP1, GAP3, FBA3, SBP1, RPE1, RPI1 and PRK1 proteins were located in the plastid stroma and not in the pyrenoid. While the two isoforms of GAPDH (GAP1 and GAP3) that are predicted to be localized to the chloroplast, indeed showed a strong Venus-signal in the stroma of this organelle, this was not true for the two isoforms of aldolase. The Venus-signal of FBA3 was found in the stroma but the Venus-signal of FBA2 was detected in the cytosol, in particular surrounding or confining the nucleus, indicating a role unrelated to the CBC.

Our pyrenoid-enrichment protocol detected 61.8% and 21.8% of Rubisco activity in the pyrenoid-enriched fraction of cells grown under LC and HC, respectively (*Figure 2*). The Venus-signals of the RBSC1 and RBSC2 subunits in the pyrenoid were very strong and very low in the stroma of non-dividing cells (*Figure 3*). The weaker stroma signal was dispersed over a much larger volume compared to the signal of the pyrenoid. Estimations by eye are therefore very difficult. *Mackinder et al. (2016)* quantified the fluorescence signal of RBSC1 and found 79% and 32% of the signal in pyrenoids in ambient-$CO_2$-grown cells and high-$CO_2$-grown cells, respectively. The lower values from the pyrenoid-enrichment protocol compared to fluorescence analyses might be because the pyrenoids were not completely stable during the enrichment protocol. Another possibility is that pyrenoids from dividing cells, which contain less Rubisco in their pyrenoid (*Freeman Rosenzweig et al., 2017*), contributed to a higher extent to the pyrenoid-enrichment analysis than to fluorescence analyses. However, the differences between the two approaches were less than 20%, which is not large considering these are from completely different methods, different laboratories and different cultivation

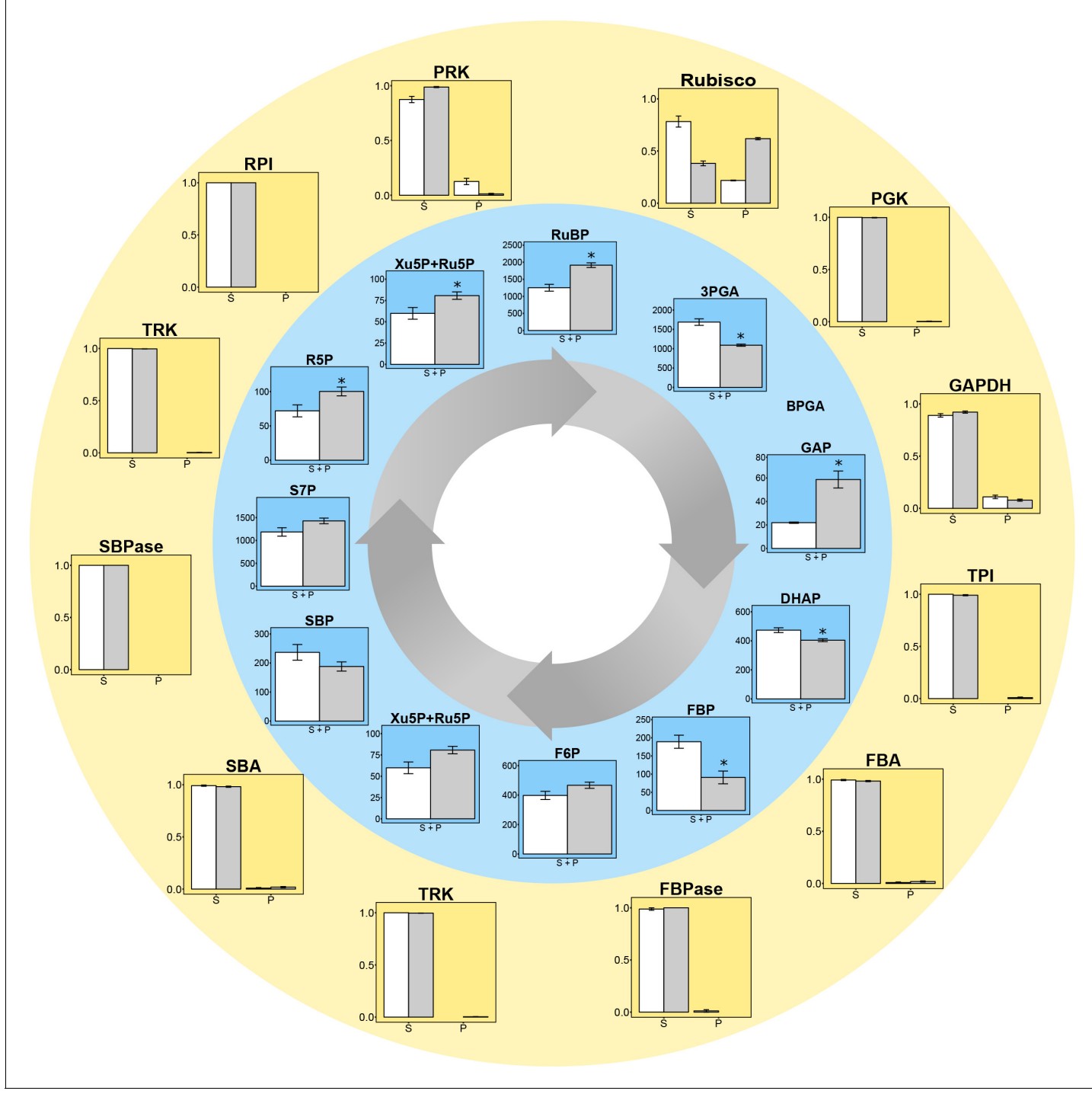

**Figure 2.** Experimental data for protein distributions (outer yellow circle) and metabolite concentrations (inner blue circle) in CCM-supressed (white bars, HC) and CCM-induced (grey bars, LC for proteins and LC* for metabolites) conditions. *Chlamydomonas reinhardtii* CC1690 cells were grown under high $CO_2$ (HC for proteins and metabolites; white bars), ambient $CO_2$ (LC for proteins; grey bars) and ambient $CO_2$ bubbled for 15 min with high $CO_2$ (LC* for metabolites; grey bars). Enzyme distribution between a pyrenoid-enriched fraction (P) and a stroma-enriched fraction (S) was determined by enzyme activity measurements (Rubisco; n = 4 ± SE) and shotgun proteomics (all other proteins; n = 4 ± SE). Metabolites of the Calvin-Benson cycle (CBC) in total cells were measured by HPLC-MS/MS. The metabolite concentrations were normalized to the chloroplast volume as described in the text and *Supplementary file 1D*, and given as absolute concentrations (µM) in the chloroplast, which includes both microcompartments, the stroma and the pyrenoid (S + P) (n = 4 ± SE). Student´s *t*-test (alpha = 0.05), significantly changed metabolites are marked with one asterisk.

The online version of this article includes the following figure supplement(s) for figure 2:

**Figure supplement 1.** Induction of carbon concentrating mechanism (CCM).

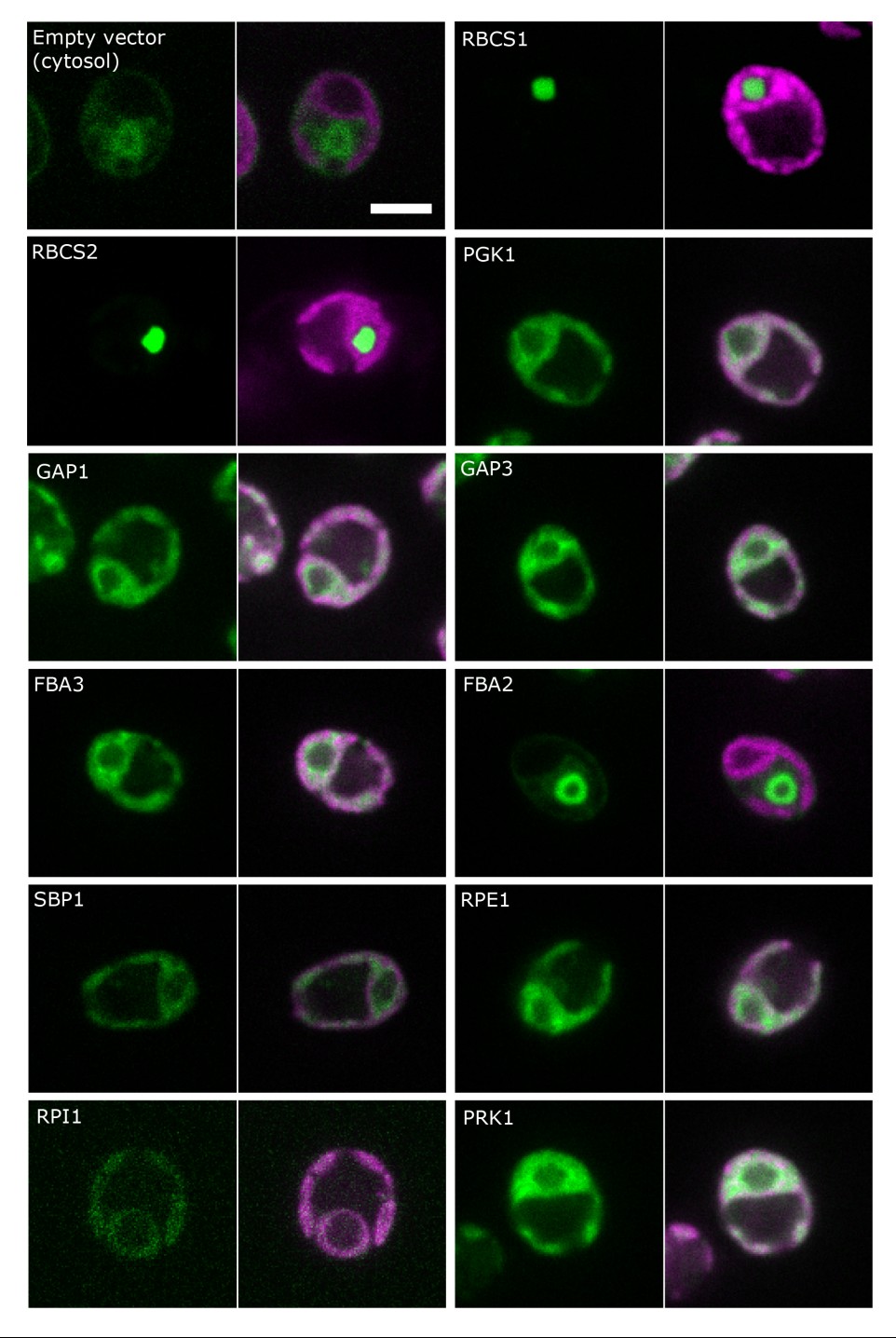

**Figure 3.** Localisation of CBC enzymes. *Chlamydomonas reinhardtii* CC-4533 cells expressing Venus-fusion constructs (green) were grown under ambient $CO_2$, imaged by fluorescence microscopy and two pictures per constructs are shown. On the left side, solely the signal of the Venus-fusion construct (green) and on the right side, the overlay picture of the signal of the Venus-fusion construct (green) and the chlorophyll fluorescence (magenta) is shown. The white bar represents 5 μm. Details on the protein names are given in the text and *Supplementary file 2*.

set-ups. For our modelling approach, we took the values of the pyrenoid-enrichment protocols shown in *Figure 2*, because metabolites were measured in the same cultivation set-up.

## Metabolites in high-CO$_2$ grown (HC) cells, and in low-CO$_2$ grown cells after 15 min of exposure to 5% CO$_2$ (LC*)

Absolute quantification of the metabolites of the CBC and starch synthesis were obtained using an ion-paired liquid-chromatography coupled to triple-quadrupole mass spectrometry (HPLC-MS/MS)-based approach (*Figure 2*, *Supplementary file 1C*). As treatments necessary to separate pyrenoid from stroma would affect metabolite levels, only the total amounts of metabolite could be quantified. To convert content per cell to concentration, whole cell volume was measured by a Coulter counter (Materials and methods), and these were normalized to the chloroplast volume (31.18% of the whole cell volume) based on 3D reconstruction of EM-stacks of *C. reinhardtii* wild-type cells (*Schötz et al., 1972*; *Supplementary file 1D*). Metabolite distributions between cytosol and chloroplast were estimated based on data obtained by non-aqueous fractionation (*Gerhardt et al., 1987*; *Supplementary file 1D*). Measurements were carried out on cells grown and harvested in high CO$_2$ (HC) and on cells grown in low CO$_2$ and bubbled with 5% CO$_2$ for 15 min before harvesting (LC*). This was done to investigate differences in metabolism between CCM-induced (LC*) and CCM-suppressed (HC) cells at comparable photosynthesis rates. This short high-CO$_2$ treatment of CCM-induced cell did not change the physiology of the cells as judged from an unchanged pyrenoid structure (*Figure 2—figure supplement 1C*). As anticipated, the rates of photosynthesis were almost equal in both sets of cells (55 and 57 µmol O$_2$ * h$^{-1}$* mg Chl$^{-1}$ for HC cells and for LC*, respectively; *Supplementary file 1E*). As this total flux remains the same between the two conditions, any differences in metabolite pools are likely due to physiological changes such as the altered distribution of Rubisco.

The LC* cells showed a significant (Student's *t*-test, p-value<0.05) decrease in 3PGA, dehydroxyactone phosphate (DHAP) and significant increases in RuBP, fructose-6-phosphate (F6P), ribose-5-phosphate (R5P), xylulose-5-phosphate (Xu5P) and ribulose-5-phosphate (Ru5P) compared to HC cells (*Figure 2*, *Supplementary file 1C*). The increased level of RuBP and increased abundance of Rubisco in the pyrenoid under LC* provided a first indication that concentration gradient to drive diffusion or transport of RuBP into the pyrenoid may be increased in LC* cells. However, it remains unclear if and how the fluxes of the reactions comprising the CBC were affected by the microcompartments. For example, the significant decrease of 3PGA under low compared to high CO$_2$ could not be readily explained without considering a systems approach in which the effects of all participating components are jointly considered. Further, interpretation of the metabolite data needs to take into consideration the altered location of Rubisco (see above).

## Modelling the effect of chloroplast microcompartmentation on the CBC

To investigate the effect of microcompartmentation on the organization and partitioning of the reaction fluxes and metabolite pools of the CBC, we devised a mathematical model tailored to *C. reinhardtii* (Materials and methods, see *Figure 4* for a graphical model representation and *Supplementary file 3* for model details). The model consists of two nominal copies of the CBC, one in the stroma and a second one in the pyrenoid, interconnected by reversible transport reactions for every CBC intermediate. We combined the model with our experimental data to ask which parts of the CBC operate in the stroma, and which in the pyrenoid. For this purpose, the volume of the stroma and the pyrenoid were needed. The volume of the stroma was determined as described above and the pyrenoid volume was calculated as the average of 12 and 10 TEM pictures of HC and LC cells of this study, respectively, applying the equation for the volume of an ellipsoid (*Supplementary file 1F*).

All enzymatic model reactions are decomposed into their elementary reactions such that the formation of the substrate-enzyme complexes is explicitly considered (Materials and methods). The resulting structure of the model was mathematically represented by the stoichiometric matrix, $N$, where rows correspond to model components (i.e., metabolites, enzymes, metabolite-enzyme complexes) and columns correspond to reactions (*Orth et al., 2010*). The entries of matrix $N$ indicated the molarity with which a component is produced (positive value) or consumed (negative value) by each reaction.

First, we sampled steady-state flux distributions, $v(c, k)$, from the flux cone, $C(N)$, given by $C(N) = \{v(c, k) | N \cdot v(c, k) = 0$ and for every reaction $R_i, v_i(c, k) \geq 0\}$ of the modeled system under the additional constraint of constant CO$_2$ uptake (398 µM/s ), ferredoxin-NADP+ reductase

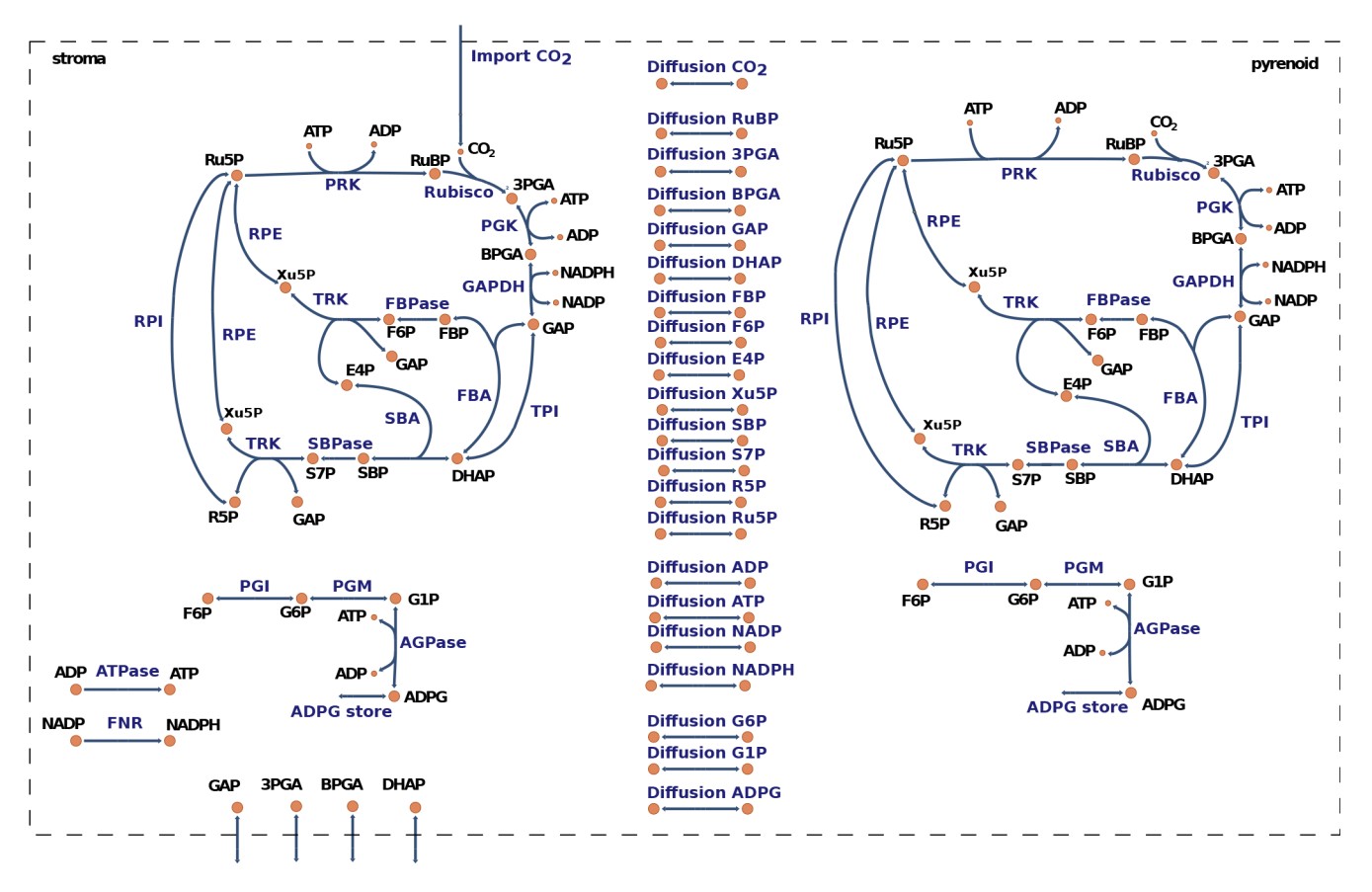

**Figure 4.** Graphical representation of the model for carbon fixation in *Chlamydomonas reinhardtii*. The model includes a copy of the Calvin-Benson cycle (CBC) in the chloroplast stroma and in the pyrenoid. In addition, the model considers reversible transport between the stroma and the pyrenoid for all CBC intermediates. A complete list of enzyme and metabolite names corresponding to the given abbreviations is presented in *Supplementary file 3*.

The online version of this article includes the following figure supplement(s) for figure 4:

**Figure supplement 1.** Distribution of Chi-square statistic for 5000 sampled steady-state flux distributions under three different assumptions.

(FNR, 796 µM/s) and ATPase activity (1.194 µM/s representing invariant NADPH and ATP formation. In addition, we integrated protein abundances from our experimental data on protein distribution in the chloroplast for HC and LC cells to draw conclusions on the physiologically relevant part of the flux cone (see Materials and methods) and to reduce the number of parameters which need to be estimated. The data were integrated under the assumption that the ratio of fluxes of the corresponding reactions between the stroma and the pyrenoid were bounded by the ratio of the experimentally-determined protein abundance (see Materials and methods).

Next, we described the flux through each elementary reaction, $v$, by the ubiquitous mass action kinetics (*Voit et al., 2015*). According to this kinetic law, the flux through reaction $R_i$ was expressed by $v_i = k_i \Pi_j c_j^{\alpha_{ji}}$, with $\alpha_{ji}$ denoting the stoichiometry with which a component (i.e. enzyme or metabolite) $j$, of concentration $c_j$, enters reaction $i$ as a substrate and $k_i$ denoted the reaction rate constant. Log-transformation of this equation yielded a system of linear equations including the log-transformed fluxes, concentrations, and rate constants. Given a steady-state flux distribution sampled from $C(N)$ the linear system could be solved using a constraint-based optimization approach (see Materials and methods) to obtain model estimates of metabolite concentrations and rate constants under the constraint that the rate constants between stroma and pyrenoid were close to each other. For reactions with known rate constants, obtained from literature, the optimization program attempted to reduce the difference between modelled $k_i$ and the respective literature values

(*Supplementary file 4A*). This approach facilitated the investigation of multiple possible modes of action of the CBC based on complete sets of model parameters. The resulting model predictions provided information about: (i) the flux state most compatible to data, (ii) microcompartment-specific metabolite and enzyme concentrations and (iii) rate constants serving as proxies for enzyme turnover rates $k_{cat}$.

Due to the ambiguous experimental data, with protein abundances indicating a low but detectable GAPDH and, sometimes, PRK activity in the pyrenoid fraction but low or non-detectable Venus-signal in the pyrenoid (*Figure 2* and *3*), we inspected model predictions for three different scenarios. We considered: (*i*) activity of PRK, Rubisco and GAPDH, (*ii*) activity of PRK and Rubisco, and (*iii*) only activity of Rubisco in the pyrenoid. To further reduce the number of parameters which need to be estimated, exchange was only allowed for metabolites involved in reactions catalyzed by enzymes present in the pyrenoid. To identify the best model, we used the Chi-square statistic, $X^2$, between modelled and measured total concentration for 11 metabolites over 5,000 steady-state flux distributions (*Figure 4—figure supplement 1*). In line with the current understanding of the CCM, where apart from Rubisco, the remaining enzymes of the CBC are hypothesized to be situated in the stroma, a significant value for $X^2$ was obtained for the scenario in which Rubisco was the only active enzyme in the pyrenoid (average Rubisco localization in pyrenoid 61.8% for LC and 21.8% for HC conditions, $\bar{X}^2_{LC*} = 2.20$, $\bar{X}^2_{HC} = 0.52$). A significant value was also observed when the activities of PRK (average PRK localization in pyrenoid 1.2% for LC and 12.6% for HC conditions), Rubisco and GAPDH (average GAPDH localization in pyrenoid 7.8% for LC and 11% for HC conditions) were allowed ($\bar{X}^2_{LC*} = 0.26$, $\bar{X}^2_{HC} = 0.31$). Simulations of this latter scenario showed no activity of GAPDH in the pyrenoid, hence the good fit in the $X^2$ statistic was a result of allowing a circular transport of NADP, NADPH, BPGA and GAP. Circular transport of metabolites between stroma and pyrenoid, without metabolites being used in the pyrenoids, however, is physiologically unlikely. Under the assumption that PRK and Rubisco were active in the pyrenoid, we observed average $X^2$ values of $\bar{X}^2_{LC*} = 74.15$, $\bar{X}^2_{HC} = 16.51$ (*Figure 4—figure supplement 1*). Hence, in the following, we only provide modelling results for the scenario with Rubisco as the only active enzyme in the pyrenoid.

Since we obtained at least 1000 samples with a significant fit between the modelled and measured concentrations under both experimental conditions, the presented findings rely on the parameter sets leading to the 1,000 best fits. In addition, we used the qualitative Pearson correlation coefficient to validate our predictions. We found that modelled quantities were in qualitatively excellent agreement with measurements under both conditions (Pearson correlation coefficient $\geq$ 0.99, p-value $< 10^{-6}$, *Figure 5*). Therefore, we further used this model to investigate and understand the influence of microcompartmentation on the function of the CBC.

## Two modes of CBC operation

Next, we investigated differences in thermodynamic characteristics in HC and LC* cells. We used the modelled metabolite concentrations and equilibrium constants, $K_{eq}$, obtained from eQuilibrator (*Flamholz et al., 2012*) to estimate $\Delta G = -RT * (ln\ K_{eq} - ln\ Q)$ for each reaction across the sampled flux distributions, where $Q$ is the ratio of active product concentrations and active reactant concentrations. In LC* cells (CCM-induced) we found TRK to be the only enzyme operating in both directions (*Figure 5—figure supplement 1*). For the remaining enzymes, except RPE and RPI for which $\Delta G > 0$, the reactions were predicted to be exergonic (i.e., $\Delta G < 0$,). The positive $\Delta G$ for reactions RPE and RPI may point to a substrate channeling (*Chiappino-Pepe et al., 2017*). We observed a change in sign of $\Delta G$ between the two conditions for enzyme TPI only (*Figure 5—figure supplement 1*).

In addition, we studied the function of the CBC at the level of reaction fluxes. We compared the net flux, corresponding to the sum of fluxes of forward and backward reaction in case of reversible reactions for the HC and LC* cells (see *Figure 6A* for the fold-changes between the HC and LC* fluxes, see *Figure 6—figure supplement 1* for the fluxes at HC and LC* separately, *Supplementary file 5* and *6*). Under the assumption that the rate of $CO_2$ uptake from the environment was the same in HC and LC* cells, the modelled rate of $CO_2$ diffusion into the pyrenoid for LC* cells showed a three-fold increase in comparison to HC cells, indicating that the model captures the function of the CCM (*Figure 6B*). The rate of the Rubisco reaction in the pyrenoid followed the $CO_2$ import into the microcompartment and was, therefore, three-fold increased under LC*

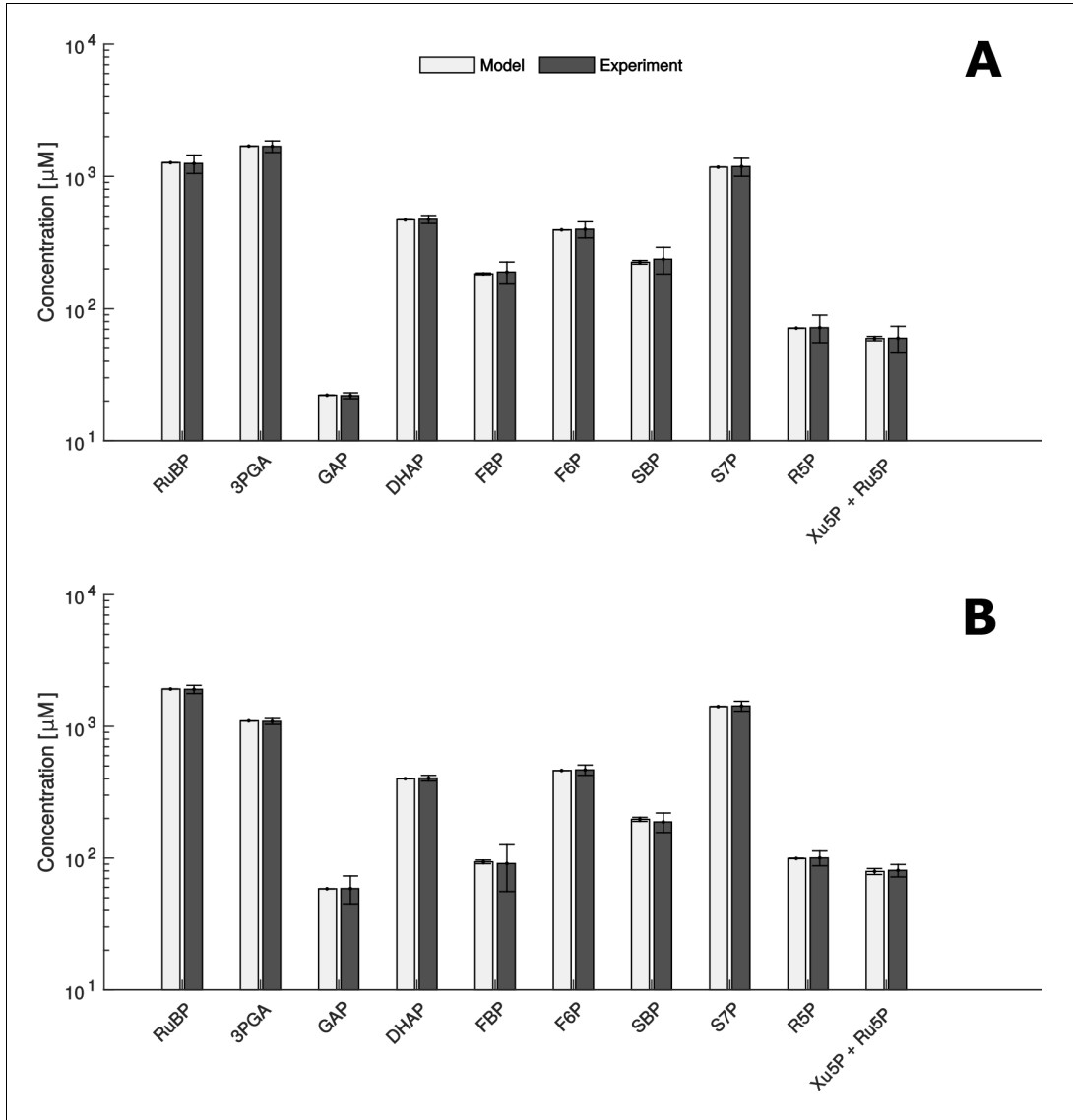

**Figure 5.** Comparison of metabolite data determined experimentally (Experiment) and by mathematical modelling (Model) under HC (**A**) and LC* (**B**) conditions. (A + B) Modelled data is shown as white bars (n = 1,000 ± SD) and the SD-values are too small to be seen. Experimentally, metabolites were measured by HPLC-MS/MS as already shown in *Figure 2* (n = 4 ± SD) (grey bars).

The online version of this article includes the following figure supplement(s) for figure 5:

**Figure supplement 1.** Distribution of estimated $\Delta G$ values for CBC metabolites over 1000 sampled steady-state flux distributions under no-CCM (HC; **A**) and CCM-induced (LC*; **B**) conditions.

conditions. Since the function of Rubisco depends not only on the import of $CO_2$, but also on the availability of RuBP, its predicted rate of import into the pyrenoid from the stroma was also three-fold increased in LC* cells. In addition, the rate of 3PGA export from the pyrenoid into the stroma was higher under LC* conditions than in HC conditions.

## Mechanisms of metabolite transport between microcompartments

To examine the mode of transport between pyrenoid and stroma, the bound and free metabolite levels were determined for both conditions (*Figure 6C*, *Supplementary file 5B* and *6B*). Due to elementary reactions considered in modelling (Materials and methods), we modelled concentrations for free metabolites as well as concentrations for the respective metabolite-enzyme-complexes. In case

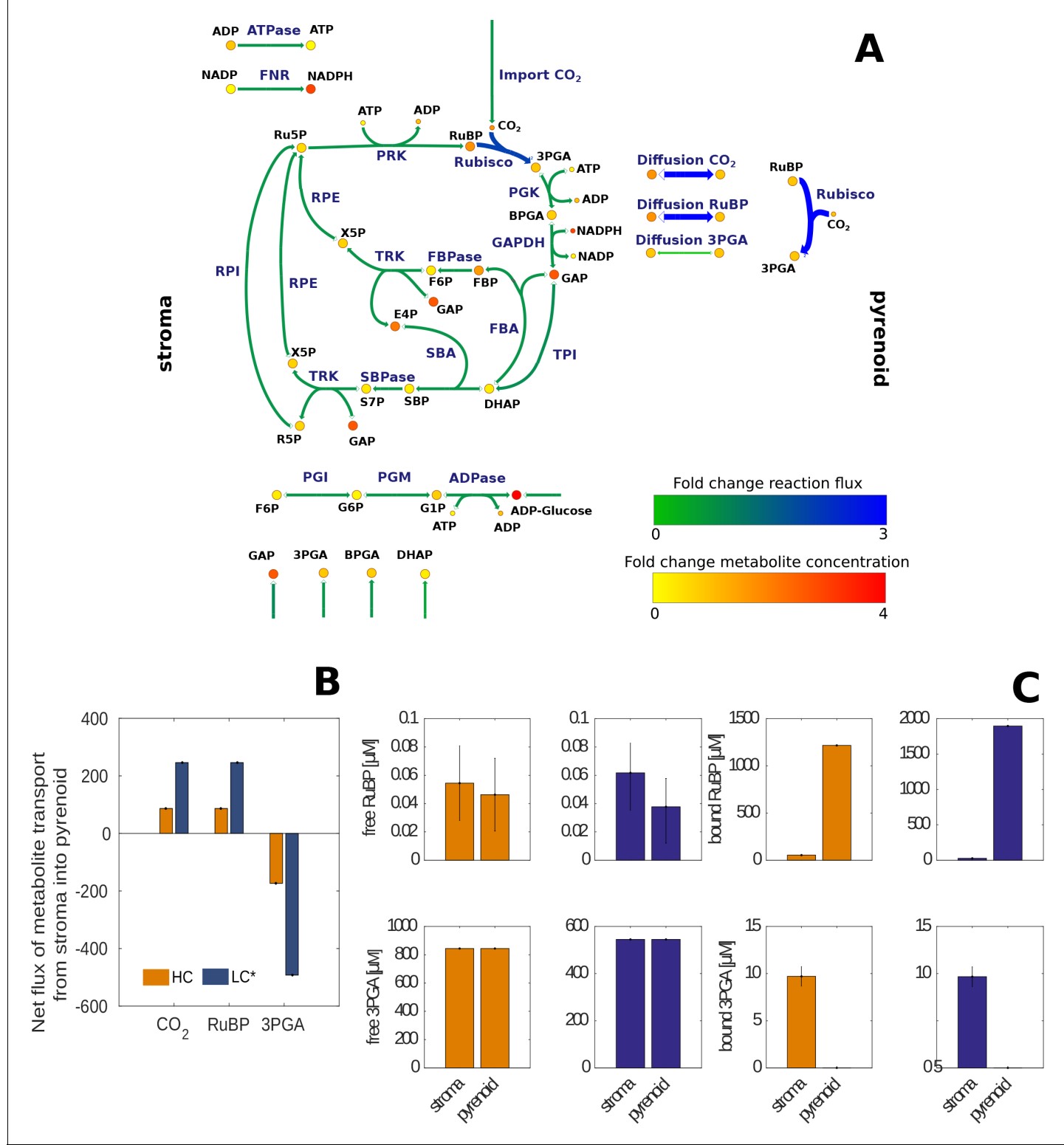

**Figure 6.** Changes in estimated reaction fluxes and metabolite concentrations for HC and LC* *Chlamydomonas reinhardtii* cells. (**A**) Fold changes of model predicted average net flux (represented by arrows) and total metabolite concentrations (represented by circles) between LC* and HC cells indicated by a colour code (see legend) and size of the arrows. The actual values are provided in *Supplementary file 5* and *6* and summarized in *Figure 6—source data 1*. The main difference observed between HC and LC* conditions was an increased flux through Rubisco in the pyrenoid and an increased flux of RuBP and 3PGA to the pyrenoid and from the pyrenoid, respectively. The flux through the Calvin-Benson cycle located in the stroma, however, is similar under both conditions (fold change of 1). (**B**) Net flux for transport of $CO_2$, RuBP and 3PGA between stroma and pyrenoid under HC

*Figure 6 continued on next page*

*Figure 6 continued*

(orange) and LC* (blue) conditions. A positive value indicates transport from stroma to pyrenoid, while a negative value indicates transport from pyrenoid into stroma. (**C**) Concentrations of bound and free RuBP and 3PGA under HC (orange) and LC* (blue) conditions.

The online version of this article includes the following source data and figure supplement(s) for figure 6:

**Source data 1.** Summary of the most important concentrations and fluxes.

**Figure supplement 1.** Reaction flux estimated for *Chlamydomonas reinhardtii* cells grown under non-CCM (HC; **A**) and CCM-induced (LC*; **B**) conditions.

a metabolite can bind multiple enzymes, the concentration of the bound metabolite was given by the sum of concentrations over the respective enzymes. The model predicted that the vast majority of the RuBP was bound. The observed increase in RuBP concentration for LC* cells was therefore mainly due to an increase in bound RuBP in the pyrenoid as a result of an increased Rubisco concentration (*Figure 6—source data 1*). The model predicted slightly higher free RuBP in the stroma in comparison to the pyrenoid (0.024 µM and 0.008 µM difference under LC* and HC conditions, respectively). Furthermore, in line with increased transport RuBP rates into the pyrenoid, the difference between free RuBP in the stroma and pyrenoid was larger in LC* than HC cells. Since in both cell types we observed a concentration gradient in the direction of RuBP transport towards the pyrenoid, the model predictions indicate diffusion or facilitated transport may be a feasible mechanism of RuBP transport under HC and LC* (although the estimated concentration gradient is very small). In contrast, the model predicted equal free amounts of 3PGA between pyrenoid and stroma under HC and LC*, implying diffusional equilibrium.

## Discussion

Quantitative and qualitative experimental data were obtained for the distribution of CBC and starch synthesis pathway enzymes between the chloroplast stroma and the microcompartment pyrenoid, confirming that Rubisco is largely located in the pyrenoids (average Rubisco localization in pyrenoid 61.8% for LC and 21.8% for HC conditions). All other CBC enzymes were present at only very low amounts (PRK, GAPDH) or were totally absent from the pyrenoid in LC cells with an operational CCM. Metabolite data measured in whole cells revealed that LC* cells have altered CBC metabolite levels to those in HC cells at the same $CO_2$ concentration and net rate of photosynthesis. However, this data alone is not sufficient to understand flux between the stroma and pyrenoid.

A kinetic model, parametrizing each reaction in the stroma and pyrenoid, allowed us to calculate flux distributions under the two distinct physiological states. The validation of the model indicated statistically significant quantitative and qualitative agreement between the experimental and modelled chloroplast metabolite concentrations. The fit for the model as a whole was assessed, considering the contribution of the predictions for each metabolite level. The agreement between experimental data and predicted metabolite levels was statistically acceptable for 10 out of 16 individual metabolites. In the case of GAP and SBP, the lack of statistical fit could be due to the fact that these compounds are involved in redox-regulated reactions, namely the reduction of BPGA to GAP catalyzed by GADPH (*Sparla et al., 2002*) and the hydrolysis of SBP catalyzed by SBPase (*Dunford et al., 1998*). The values given by the model are lower than the experimental pool sizes implying that there might be incomplete redox activation. This discrepancy is in line with the fact that the model does not include redox regulation. This explanation is supported by the observation that the applied light intensity of 46 µmol photons$*m^{-2}*s^{-1}$ is not saturating, and analysis of SBP and S7P indicated that SBPase is incompletely activated at this light intensity (*Mettler et al., 2014*). For the reaction catalyzed by GAPDH, it is likely that NADPH provided by the light reaction was rate limiting rather than GAPDH redox activation (*Mettler et al., 2014*).

To provide further validation of the model, we compared the predicted rate constants with values for $k_{cat}$ available from literature that were not used in the model parameterization (Materials and methods; *Supplementary file 4A*). This was the case for TRK (Xu5P, rate constant = 37 $* s^{-1}$; Ru5P, rate constant = 37 $* s^{-1}$) (*Supplementary file 4B*) for which the published $k_{cat}$ values in spinach are $<0.02 * s^{-1}$ for Xu5P and Ru5P (*Teige et al., 1998*). The values for the plant TRK reported in literature are surprisingly low, since those in yeast ($k_{cat} = 113 * s^{-1}$) and human ($k_{cat} = 9 * s^{-1}$) (*Albe et al., 1990*) are much higher.

The first important output of the model included the fluxes of the exchange reactions between the pyrenoid and the stroma. The mathematical model enabled us to calculate exchange fluxes between chloroplast stroma and pyrenoid based on enzyme partitioning, rate constants of enzymatic reactions and steady-state metabolite levels. The model predicted an increase in the flux of the Rubisco reaction inside the pyrenoid under LC* compared to HC conditions. This prediction reflects the increased presence of Rubisco in the pyrenoid under these conditions. The model also predicted import of RuBP and export of 3PGA into and from pyrenoid, respectively, with both fluxes being higher under LC* compared to HC conditions. The concentration gradient for RuBP between stroma and pyrenoid was larger under LC* than under HC, but for both conditions the concentration gradients were small (0.024 µM and 0.008 µM, respectively). In case of 3PGA, the model predicted, for both conditions, that there is no concentration gradient between stroma and pyrenoid. Since there is no accumulation of RuBP in the pyrenoid or 3PGA in the stroma, active transport seems unnecessary and movement between the microcompartments may occur by diffusion.

Previously, the starch sheath surrounding the pyrenoid under ambient but not under high $CO_2$ conditions, was suggested to work as diffusional barrier to prevent $CO_2$ from diffusing out of the pyrenoid (*Badger and Price, 1994*). In this case, the starch sheath would also represent a diffusional barrier for RuBP entering into the pyrenoid. In the model, such an increased diffusional barrier could be seen as a change in the diffusion constant between HC, where no starch sheet was present, and LC*, where a starch sheet was established. The diffusion constants is proportional to 10.623 s$^{-1}$ (86.664 µM s$^{-1}$ * 0.008158$^{-1}$ µM$^{-1}$) and 10,259 s$^{-1}$ (246.12 µMs$^{-1}$ * 0.02399$^{-1}$ µM$^{-1}$) for HC and LC*, respectively (*Figure 6A*, *Figure 6—source data 1*), revealin *Figure 6—figure supplement 1g* that the difference in the diffusion constants was minor. Therefore, our model does not support the idea of an increased diffusional barrier between pyrenoid and stroma under LC* for RuBP. This finding is experimentally supported by the fact that a starch-less mutant, also lacking the starch sheath around the pyrenoid, is still able to develop a fully functional CCM (*Villarejo et al., 1996*).

As the fluxes of both 3PGA and RuBP are increased under LC* compared to HC with minor (RuBP) or no (3PGA) increase in the concentration gradients, flux is increasing with minor or no change in the driving force. This implies that instead of a strong diffusional boundary there may be facilitated diffusion under LC* compared to HC. Recently, several proteins that are known to be expressed under low $CO_2$ conditions but are of unknown function were sub-cellularly localized (*Mackinder et al., 2017*). LCIC/LCIB and LCI9 are expressed in the proximity of the starch sheath and surrounding the pyrenoid, which is consistent with one or several of these proteins being involved in facilitating diffusion of RuBP into and of 3PGA out of the pyrenoid. This would occur exclusively under low $CO_2$ conditions, as the according genes are only expressed under low $CO_2$ (*Yamano et al., 2008*; *Yamano et al., 2010*). Recently it was suggested that LCIC/LCIB most likely has a carbonic anhydrase activity (*Jin et al., 2016*), but additional functions are still possible.

A high resolution cryoEM study of the *C. reinhardtii* chloroplast (*Engel et al., 2015*) revealed the presence of pyrenoid minitubules that form narrow continuous channels between the inter-thylakoid stromal space and the pyrenoid matrix. One function of these minitubules could be to facilitate diffusion of RuBP and 3PGA, apart or in addition to the protein candidates mentioned above. The estimated minitubule lumen diameter is of the order of 3–4 nm by 8–15 nm, which is not much larger than the longest axes of RuBP and 3PGA (about 1.2 nm and 0.7 nm, respectively), so diffusion of these metabolites in the minitubules would be possible. It will be interesting to learn if the physiochemical properties of these minitubules favor facilitate diffusion of these metabolites, either as free acids or as the magnesium complexes that are likely to predominate at pH and magnesium concentration found in the stroma in the light (*Portis, 1981*; *Werdan et al., 1975*). Charge properties might, speculatively, provide a means to discriminate between CBC intermediates and the weakly anionic bicarbonate and neutral $CO_2$.

Under low $CO_2$ conditions and a fully functional CCM, the limiting Rubisco substrate $CO_2$ is concentrated in the pyrenoid. Moreover, our data indicate that the CCM also establishes structures that allow facilitated transport of RuBP, the other Rubisco substrate, from the stroma to the pyrenoid and the release of the Rubisco product 3PGA into the stroma, where the rest of the CBC enzymes are located. Such channels for exchange of CBC metabolites were suggested to be present in the pyrenoid of *C. reinhardtii* (*Engel et al., 2015*) and the proteinaceous shell of the carboxysome based on the number of shell proteins and their localization in the shell (*Kerfeld and Melnicki, 2016*). However, supporting experimental data to our knowledge is scarce so far and therefore our study is the

first with underlying experimental metabolite data postulating such transport reactions in a carbon-concentrating microcompartment.

Altogether, our systems biology approach allowed us to demonstrate (*i*) that changes in microcompartments cause specific inhomogeneities that affect steady-state metabolite levels and have to be considered in mathematical modelling approaches based on such experimental data; (*ii*) that mathematical models with mild assumptions can be used to study flux distributions between reactions inside and outside microcompartments, which are very difficult and often technically impossible to study experimentally; and (*iii*) that metabolites can be identified that are exchanged between the compartments and their exchange fluxes quantified. Our study opens the possibility to investigate the effects of microcompartmentation in different cellular scenarios and to understand their role in the overall physiology of the investigated system.

# Materials and methods

## Key resources table

| Reagent type (species) or resource | Designation | Source or reference | Identifiers | Additional information |
|---|---|---|---|---|
| Strain, strain background (*Chlamydomonas reinhardtii*) | CC1690 wild-type strain | Chlamydomonas Resource Center | RRID: SCR_014960 | |
| Strain, strain background (*Chlamydomonas reinhardtii*) | CC-4533 wild-type strain | Chlamydomonas Resource Center | RRID: SCR_014960 | - |
| Antibody | rabbit Anti-Beta-CA1 | AgriSera | Cat# AS11 1737; RRID:AB_10752086 | (1:7500) |
| Antibody | rabbit Anti-AtpD | AgriSera | Cat# AS10 1590; RRID:AB_10754669 | (1:30000) |
| Software, algorithm | MaxQuant | MaxQuant (http://www.biochem.mpgde/5111795/maxquant) | RRID:SCR_014485 | version 1.5.2.8 |
| Software, algorithm | Codes used for modelling | | | The mathematical models can be accessed via https://github.com/ankueken/Chlamy_model (copy archived at https://github.com/elifesciences-publications/Chlamy_model). |

## Cell growth

*Chlamydomonas reinhardtii* CC1690 wild-type strain (*Sager, 1955*) was obtained from the Chlamydomonas Resource Center (RRID:SCR_014960) and cultivated as described in *Mettler et al. (2014)*. Cells were growth photoautotrophically in a 5-litre bioreactor BIOSTATB-DCU (Sartorius Stedim, Germany) for five days until a cell density of $3-5*10^6$ cells*$ml^{-1}$ was reached and thereafter, turbidity was kept constant. The culture in the bioreactor was constantly stirred with 50 rpm at 24°C, exposed to an average of 46 μmol photons*$m^{-2}*s^{-1}$ (measured internally at four different positions) and bubbled at a rate of 400 ccm with air enriched with 5% $CO_2$. Turbidity was kept constant by medium exchange (125 ml*$h^{-1}$) for two days before harvesting the high-$CO_2$-grown cells. Then the air bubbling of the bioreactor was changed from 5% $CO_2$ to ambient air. Cells were adapted to the low-$CO_2$ conditions for 30 hr. Turbidity was kept constant by medium exchange (39 ml*$h^{-1}$) and $CO_2$ in the outlet air of the bioreactor was measured (*Figure 2—figure supplement 1*). The $CO_2$ level

measured by gas chromatography in the outlet air of the bioreactor dropped within 12 hr from 4.5% (bubbling with 5%) to a constant 0.02% (bubbling with air of approximately 0.039% $CO_2$). Levels inducing CCM (0.1%) were already reached after 4 hr but the system needed at least an additional 8 hr to equilibrate at 0.02% $CO_2$. We can therefore assume that cells in the bioreactor are exposed to, at a maximum, 0.02% $CO_2$. LC cells were harvested after 30 hr of bubbling with ambient air and LC* cell after an additional 15 min bubbling with air enriched with 5% $CO_2$. Cell number and cell volume were determined by a Z2 Cell Coulter (Beckman Coulter, USA) in triplicates of 100-fold diluted samples.

## Cell harvesting

Before harvesting, 500 ml of, both, the high- and the low-$CO_2$-grown cells were transferred to a 1 L glass bottle and kept at the same light intensity (46 µmol photons*$m^{-2}$*$s^{-1}$, measured internally as indicated above) as in the bioreactor, stirred and bubbled with ambient (LC) or 5% $CO_2$ (HC and LC*) for 15 min. For metabolite measurement, the cells were quenched with 70% cold methanol as described in *Mettler et al. (2014)*. For enzyme activities and proteomics experiments, 10 ml of cells were spun-down at 4000 rpm for 2 min at 4°C and stored at −80°C before usage.

## Metabolite measurements by HPLC-MS/MS

Polar metabolites were extracted with chloroform/methanol/water, separated with an ion-paired liquid chromatography and detected on a triple quadrupole as described in *Mettler et al. (2014)*. The absolute amounts of metabolites measured by HPLC-MS/MS were normalized to the cell volume determined by a Coulter counter described above. The metabolite levels were then normalized to the chloroplast volume according to *Schötz et al., 1972* using distributions between cytosol and chloroplast according to *Gerhardt et al. (1987)*. See *Supplementary file 1D* and the Result section for more details.

## Enzymatic activities

The ten ml algal material was defrosted on ice and extracted with extraction buffer (EB) containing 2% Triton (50 mM HEPES, 20 µM leupeptin, 500 µM DTT, 1 mM PMSF, 17.4% glycerol). The samples were sonicated 3 × 15 s (six cycles, 50% intensity, Sonoplus Bandelin electronics, Germany) and kept on ice in between for 90 s. Half of the sample was then used for analysis of total enzyme activity and half was centrifuged at 14000 rpm for 2 min to obtain a soluble and pellet fraction. The pellet was washed twice with 500 µl EB before resuspension in 500 µl EB. The enzyme activities of total, soluble and pellet fraction were analyzed together in 96 well microplates using a Janus pipetting robot (Perkin-Elmer, Belgium) and absorbances were determined using a Synergy, an ELX-800 or an ELX-808 microplate reader (Bio-Tek, Germany). For each enzyme three different dilutions of each algal sample were measured (final concentrations in assay were 1:60, 1:300, 1:600). The AGPase (*Gibon et al., 2004*), PGM (*Manjunath et al., 1998*) and Rubisco (*Sulpice et al., 2007*) enzymatic assay were performed as described previously. For Rubisco activity measurement, the assay length was adjusted to 30–60 min.

## Proteomics data

The soluble and pellet fractions described above were subjected to shotgun proteomics analysis as described in *Mackinder et al. (2016)*. Data analysis was done using the MaxQuant Software (RRID: SCR_014485; *Cox and Mann, 2008*).

## Protein localization

Proteins were tagged and sub-compartmentally localized as described in *Mackinder et al. (2017)*. Briefly, for the fluorescence protein tagging, open reading frames of CBC genes were PCR amplified (Phusion Hotstart II, Thermo Fisher Scientific, U.S.A.) from genomic DNA, gel purified and cloned in-frame with a C-terminal Venus-3xFLAG (pLM005) tag (*Mackinder et al., 2016*) by Gibson assembly. Junctions were Sanger sequenced and constructs were linearized by either EcoRV or DraI prior to transformation into WT *C. reinhardtii* (CC-4533). For transformation by electroporation, 14.5 ng kbp-1 of cut plasmid was mixed with 250 µL of 2 × $10^8$ cells $mL^{-1}$ at 16°C in a 0.4 cm gap electroporation cuvette then transformed using a Gene Pulser II (Bio-Rad Laboratories, U.S.A.) set to 800V and 25uF.

Transformed cells were selected on Tris-acetate-phosphate (TAP) paromomycin (20 µg mL-1) plates and maintained in low light (5–10 µmol photons $m^{-2}$ $s^{-1}$) until screening for fluorescence using a Typhoon Trio fluorescence scanner (GE Healthcare, U.S.A.).

For confocal microscopy, Venus-tagged lines were grown photoautotrophically in Tris-phosphate (TP) liquid medium in air (ambient $CO_2$) at 150 µmol photons $m^{-2}$ $s^{-1}$ light intensity. 15 µL of cells at ~2–4×$10^6$ cells $mL^{-1}$ were pipetted onto poly-L-lysine coated plates (Ibidi) and overlaid with 120 µL of 1% TP low-melting-point agarose at ~34°C to minimize cell movement. Images were acquired using a spinning-disk confocal microscope (Leica DMI6000, Leica Microsystems, Germany). Venus signal was detected by 514 nm excitation with 543/22 nm emission and chlorophyll using 561 nm excitation with 685/40 nm emission. Images were analyzed using Fiji (*Schindelin et al., 2012*).

## Model description

The model was constructed to simulate carbon fixation in the chloroplast under different $CO_2$ availability. The model included two compartments: (*i*) the chloroplast stroma and (*ii*) the pyrenoid, a microcompartment located inside the chloroplast associated with the operation of a CCM. The postulated function of the pyrenoid is to generate a $CO_2$-rich environment around the photosynthetic enzyme Rubisco (*Kuchitsu et al., 1988b*; *Kuchitsu et al., 1991*). During carbon fixation Rubisco catalyses the production of two molecules 3PGA from RuBP and $CO_2$. Moreover, it catalyses the first reaction of the photorespiratory pathway: the reaction of RuBP and oxygen ($O_2$) to 3PGA and 2-phosphoglycolate. The model, however, did not include this reaction since under the high $CO_2$ conditions, this reaction was very likely suppressed (*Supplementary file 1E*).

To investigate the role and interplay of the pyrenoid and the CBC we included two full copies of the CBC, one in the chloroplast stroma and one in the pyrenoid. The copies of the CBC were linked by reversible transport reactions for all CBC intermediates. The enrichment of $CO_2$ in the pyrenoid (CCM) can be achieved via diffusion of $CO_2$ from the stroma into the pyrenoid. A full list of model reactions and components is presented in *Supplementary file 2*.

## Model construction

The model simulated carbon fixation on the chloroplast level and included 62 reactions distributed over the chloroplast stroma and the pyrenoid. To compare the simulated data and experimentally determined enzyme parameters, each reaction was modelled by its elementary reaction steps. More specifically, given an irreversible reaction $A \rightarrow B$ catalyzed by enzyme $E$, the model included three elementary reactions: $A + E \leftrightarrow AE$ and $AE \rightarrow B + E$. For reversible reactions $C \leftrightarrow D$ the model includes six elementary reactions $C + E \leftrightarrow CE$, $CE \rightarrow D + E$, $D + E \leftrightarrow DE$ and $DE \rightarrow C + E$.

After splitting of reactions the model consists of 226 irreversible reactions and 128 components (metabolites, enzymes, enzyme-substrate complexes). Each CBC copy, thereby, comprised 65 elementary reactions linked by 40 irreversible diffusion reactions. Moreover, the model included eight irreversible reactions transporting triose-phosphates from the stroma into the cytosol and vice versa, ATPase reaction converting ADP to ATP and a simplified ferredoxin-NADP + reductase (FNR) reaction converting NADP to NADPH. The production of NADPH and ATP were calculated from the measured production of oxygen, assuming two molecules of NADPH produced per oxygen molecules and 1.5 ATP molecules per NADPH molecule (*Supplementary file 1G*).

The underlying system of ordinary differential equations (ODE), therefore, simulated a system of $n = 226$ reactions acting on $m = 128$ model species and was formulated as $\frac{dc_i}{dt} = \sum_{j=1}^{m} n_{ij} v_j$, where $v_j$ was the flux trough reaction $j$ and $n_{ij}$ the respective stoichiometric coefficient in the stoichiometric matrix $N$, indicating the molarity with which substrate $i$ enters reaction $j$. The ODE system was used to simulate the steady-state concentrations $c$ of all model species under different environmental conditions by solving $\frac{dc_i}{dt} = \sum_{j=1}^{m} n_{ij} v_j = 0$. The reaction flux $v(c, k)$ depended on species concentrations $c$ and reaction rate constants, $k$, and is calculated using the law of mass-action. Thus, the flux through an irreversible reaction $j$ was given by $v_j(c,k) = k_j \prod_i c_i^{n_{ij}^-}$, where $n_{ij}^- = -n_{ij}$ if $n_{ij} < 0$ and 0, otherwise.

## Model parameterization

Here, we describe the procedure of determining parameter values for $c$ and $k$ for sampled flux distributions, $v$.

To guarantee a steady state, we first sampled steady-state flux distributions with the COBRA Toolbox (*Schellenberger et al., 2011*) function *sampleCbModel* by sampling solutions of the linear program in (*Equation 1*). To obtain flux distributions leading to a high quality fit between simulated and measured data and to reduce the number of parameters which need to estimated, we integrated the measured enzyme distribution between stroma and pyrenoid, whereby the flux ratio between pyrenoid and stroma for a reaction catalyzed by enzyme $E$ follows the measured ratio of protein abundance. Therefore, we sampled flux distributions from the solution space of the following linear program:

$$Nv = 0$$
$$v_{pyrenoid} = q\left(v_{stroma} + v_{pyrenoid}\right)$$
$$v^{FNR} = 796$$
$$v^{ATPase} = 1.194$$
$$v^{CO2\ uptake} = 398$$
$$0 \leq v \leq v_{max}$$

(1)

Consequently, if $e_{pyrenoid} = q(e_{stroma} + e_{pyrenoid})$, where $e$ was the measured relative enzyme activity or if available amount, then $v_{pyrenoid} = q(v_{stroma} + v_{pyrenoid})$. While $q$ was chosen uniformly at random for each sampled flux distribution from the measured range. For enzymes with measured relative activity in the pyrenoid below 5%, we considered only the enzyme located in the stroma to be active. In addition, based on experimental data we fixed the rate of $CO_2$ uptake to $398 \sim \mu M/s$. As upper limit of flux through a reaction we used $v_{max} = 1,000 \sim \mu M/s$. Moreover, we fixed the flux through RNR and ATPase to constant values estimated from experimental data since light intensity was unaltered.

For each flux distribution $v^*$ obtained from sampling, we determined species concentrations $c$ and reaction rate constants $k$ leading to the respective flux distribution under mass-action kinetics by solving the program in *Equation 2*:

$$\min \varepsilon^+ + \varepsilon^-$$

$$\forall_{j=1:n}$$

(2)

$$\log v_j^* = \log k_j + \sum_{i=1}^m n_{ij}^- \log c_i$$

$$\log k_{pyrenoid} - \log k_{stroma} + \varepsilon^+ = 0$$

$$\log k_{pyrenoid} - \log k_{stroma} - \varepsilon^- = 0$$

$$\log k_{min} \leq \log k \leq \log k_{max}$$

$$\log c_{min} \leq \log c \leq \log c_{max}$$

$$\varepsilon^+ \geq 0, \varepsilon^- \geq 0.$$

The lower bounds for the free concentrations of metabolites other than RuBP, BPGA and 3PGA in the linear program in Equation 2 were set to 90 and 99% to ensure agreement with the experimental data on the total metabolite levels. The lower bound for 3PGA were set to 50% of measured total concentration, while for RuBP and BPGA generic lower bounds of $c_{min_i} = 0.01$ μM were used. The upper bounds for all free metabolite concentrations were set to 100% of the measured total concentration.

Since we assumed that $k$ depends not only on environmental parameter, like temperature and pressure, but also includes regulation, which cannot be integrated in mass-action kinetics (e.g., allosteric regulation), we asked for the minimum difference in $k$ between pyrenoid and stroma leading to a feasible solution of the program. For reactions with known enzymatic turnover obtained from BRENDA (*Chang et al., 2015*; *Supplementary file 3*), we integrated this information by restricting the respective parameter boundaries. Since the model provides subcompartment-specific estimates of metabolite concentrations, we considered the sum of the respective stroma and pyrenoid metabolite concentrations and compared them to the estimated and measured metabolite concentrations in the chloroplast. We then determined the goodness-of-fit for each set of simulated metabolite concentrations and enzyme distribution and rank the parameter sets based on their chi-square value considering the 1,000 top ranked for further investigation.

## Net flux calculation

In case of non-enzymatic reactions of form $A \underset{v_b}{\overset{v_f}{\rightleftarrows}} B$ the reported net flux was calculated as $v_f - v_b$.

Since the activity of an enzyme is given by the rate of product formation per unit of time, we consider reaction flux $v_{cat}$ as net flux for an irreversible reaction catalyzed by enzyme $E$ with elementary reactions $A + E \underset{v_b}{\overset{v_f}{\rightleftarrows}} AE$ and $AE \overset{v_{cat}}{\rightarrow} B + E$. In case of reversible enzymatic reactions modelled by elementary reactions $C + E \underset{v_{b1}}{\overset{v_{f1}}{\rightleftarrows}} CE$, $CE \overset{v_{cat\_f}}{\rightarrow} D + E$, $D + E \underset{v_{b2}}{\overset{v_{f2}}{\rightleftarrows}} DE$ and $DE \overset{v_{cat\_b}}{\rightarrow} C + E$, we consider $v_{cat\_f} - v_{cat\_b}$ as reported net flux.

## Acknowledgements

We thank Carola Päpke and Joost van Dongen for the usage of equipment for photosynthesis rate measurements and Marc-Aurel Schöttler and Arren Bar-Even for critical reading. AK acknowledges support by the Max Planck Society and TM-A acknowledges support by the Deutsche Forschungsgemeinschaft (EXC 1028). FS, MSt, MSch, and TM-A acknowledge support by the Federal Ministry of Education and Research (BMBF), Germany, within the frame of the GoFORSYS Research Unit for Systems Biology (FKZ0313924). MJ acknowledges support by the National Science Foundation (EF-1105617), the National Institutes of Health (DP2-GM-119137), and the Simons Foundation and HHMI (55108535).

## Additional information

### Funding

| Funder | Grant reference number | Author |
|---|---|---|
| Deutsche Forschungsgemeinschaft | EXC 1028 | Tabea Mettler-Altmann |
| Bundesministerium für Bildung und Forschung | FKZ0313924 | Frederik Sommer<br>Liliya Yaneva-Roder<br>Michael Schroda<br>Mark Stitt<br>Tabea Mettler-Altmann |
| Max-Planck-Gesellschaft | Open-access funding | Anika Küken |
| National Science Foundation | EF-1105617 | Martin C Jonikas |
| National Institutes of Health | DP2-GM-119137 | Martin C Jonikas |
| Simons Foundation | 55108535 | Martin C Jonikas |

The funders had no role in study design, data collection and interpretation, or the decision to submit the work for publication.

## Author contributions
Anika Küken, Conceptualization, Formal analysis, Validation, Investigation, Visualization, Methodology, Writing—original draft, Writing—review and editing; Frederik Sommer, Formal analysis, Investigation, Methodology, Writing—review and editing; Liliya Yaneva-Roder, Melanie Höhne, Investigation, Methodology; Luke CM Mackinder, Investigation, Visualization, Writing—review and editing; Stefan Geimer, Formal analysis, Investigation; Martin C Jonikas, Supervision, Writing—review and editing; Michael Schroda, Conceptualization, Supervision, Writing—review and editing; Mark Stitt, Conceptualization, Funding acquisition, Validation, Writing—review and editing; Zoran Nikoloski, Conceptualization, Supervision, Validation, Methodology, Writing—original draft, Writing—review and editing; Tabea Mettler-Altmann, Conceptualization, Data curation, Formal analysis, Supervision, Validation, Investigation, Visualization, Methodology, Writing—original draft, Writing—review and editing

## Author ORCIDs
Anika Küken http://orcid.org/0000-0003-1367-0719
Luke CM Mackinder https://orcid.org/0000-0003-1440-3233
Zoran Nikoloski http://orcid.org/0000-0003-2671-6763
Tabea Mettler-Altmann http://orcid.org/0000-0002-9161-4889

## Decision letter and Author response
Decision letter https://doi.org/10.7554/eLife.37960.sa1
Author response https://doi.org/10.7554/eLife.37960.sa2

# Additional files

## Supplementary files
- Supplementary file 1. Experimental data.

- Supplementary file 2. Protein information for Venus localization.

- Supplementary file 3. Model structure.

- Supplementary file 4. Predications of $k_{cat}$-values compared to literature values.

- Supplementary file 5. Raw data of the model for HC conditions.

- Supplementary file 6. Raw data of the model for LC* conditions.

- Transparent reporting form

## Data availability
All data generated or analysed during this study are included in the manuscript and supporting files. Source data files have been provided for Figures 2, 5 and 6. All modelling related code and data are available on GitHub https://github.com/ankueken/Chlamy_model (copy archived at https://github.com/elifesciences-publications/Chlamy_model).

The following datasets were generated:

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
