## [Decision Letter]

Thank you for submitting your article "Effects of microcompartmentation on flux distribution and metabolic pools in *Chlamydomonas reinhardtii* chloroplasts" for consideration by *eLife*. Your article has been reviewed by three peer reviewers, and the evaluation has been overseen by a Reviewing Editor and Ian Baldwin as the Senior Editor. The reviewers have opted to remain anonymous.

The reviewers have discussed the reviews with one another and the Reviewing Editor has drafted this decision to help you prepare a revised submission.

Summary:

This work begins the process of understanding how cellular compartmentalization is playing mediating the flux balance of a biochemical system.

Essential revisions:

1) There are issues surrounding the potential localization and quantification of this partitioning for Rubisco that need to be discussed/tested depending on how the study was conducted.

2) Better clarity on how fractional metabolite levels were estimated.

3) There was concern that the lack of allosteric factors in the model was creating the lack of fitting on specific metabolites. This potential hindrance of the model should be discussed and the potential role of this in limiting the model fit should be commented upon.

*Reviewer #1:*

The current manuscript addresses an important issue of green algal cell biology, the distribution of enzymes of the Calvin Benson cycle and metabolites within microcompartments of the chloroplast (pyrenoid, stroma) as well as their dependence on high and ambient CO_2_ concentrations. These data are the basis for modeling approaches. The model predicts that the substrate and end product of Rubisco diffuse in and out of the pyrenoid with higher fluxes in cells with ambient CO_2_ levels.

The authors measure Rubisco activity in fractions enriched in pyrenoid and stromal proteins and find differences in the activity in both fractions that change in their ratio, depending on the CO_2_ level (Figure 2). The authors write that more than 60% of Rubisco is in the pyrenoid in LC cells (ambient CO_2_ level) and about 20% in HC cells (5% CO_2_ level). Then, they perform immunolocalization studies with Venus-fusion constructs (Figure 3). To my mind, the results shown in Figure 3 contradict the former findings of the authors. The small Rubisco subunits *RBCS1* and *RBCS2* are only visible in the pyrenoid and not in the other parts of the chloroplast, including the stroma. Unfortunately, the authors do not mention if the used cells were grown under LC or HC for these experiments. In any case, a certain portion of Rubisco should be distributed over the chloroplast stroma if Rubisco is present to a significant extent in the stroma, and the results are not an artifact of an impure stromal fraction. The stromal fraction was obtained by lysis of the cells and a centrifugation step using the resulting supernatant. I am afraid that this fraction may be contaminated with pyrenoid proteins that derive from partial lysis of the pyrenoid when disrupting the cells. Figure 1A in Mackinder et al. (2016), to which the authors refer, shows parts of pyrenoids as well, being in agreement with this possibility. An alternative would be that the fusion construct is not reflecting the real localization pattern of the smaIl Rubisco subunits. I suggest that the authors clarify this important step by conducting immunolocalization experiments with cells grown at LC and HC using anti-Rubisco antibodies before they draw their final conclusions. Especially under HC, it should be visible that the Rubisco is not only present in the pyrenoid but also in the rest of the plastid.

Another critical point is that the authors change the CO_2_ treatment conditions when performing the metabolites studies. In the latter case, they use LC cells that have been treated for 15 min with 5% CO_2_ before harvesting. Their reasons are understandable, but it would be desirable to have the same conditions for the protein and metabolites studies.

In summary, this is an interesting and highly important topic and the manuscript should be of interest to a broad readership. However, the authors have to address the mentioned concerns to either verify their conclusion or to change it and consequently adapt their model.

*Reviewer #2:*

This is a very exciting manuscript detailing the distribution of many enzymes found in the chloroplast of *Chlamydomonas reinhardtii*. The authors are mainly focused on the location of the Calvin-Benson cycle (CBC) enzymes and whether a specific protein is in the pyrenoid or chloroplast stroma and whether they moved if the cells had a functioning CCM or not. They clearly demonstrate the localization by measuring enzymatic activity after fractionation, using an MS-based method on chloroplast fractions to quantitate protein abundance and using fluorescently tagged proteins to visualize their location (Figure 3). This work is solid and the conclusions are well considered.

The authors then go on to estimate the metabolite concentrations of a number of CBC intermediates and model metabolite fluxes between the stroma and pyrenoid. This is difficult to address experimentally but this work is the best estimation of the metabolite fluxes and concentrations to date. The fact that the estimated pyrenoid and stromal concentrations of RuBP and PGA are similar is very interesting.

In my opinion the authors should address two issues near the beginning of the Discussion. The most serious issue relates to the estimation of metabolite concentrations. If I understand the methods correctly, the authors have quantitated the metabolite amount in the entire cell and have not tried to determine the metabolite amounts in the chloroplast versus cytoplasm. If that is true, it is not clearly stated in the beginning of the Discussion. How were chloroplast CBC metabolite levels estimated if the measured amount of each metabolite was for the entire cell? The authors need to say exactly what assumptions they used to calculate the amount of the metabolite in the stroma. The only place this was addressed was in the legend to Figure 2. It should be in the Materials and methods section and reiterated at the beginning of the discussion.

My second concern was which value for the percentage of Rubisco in the pyrenoid was used? Clearly the 60% pyrenoid localization seen in fractionation is lower than earlier estimates using immunogold or the fluorescence measurements. Again at the beginning of the model description and in the discussion the actual values used should be stressed. From Figure 1 it would appear that the value used was 100% in the pyrenoid but I am not certain. I think any value over 90% is reasonable but the authors should state why they favored the fluorescence method or literature values over the MS or activity measurements. I suspect there is some dissolution of the pyrenoid and that is what accounts for the discrepancy but it should be noted.

*Reviewer #3:*

This manuscript examined the localisation of all enzymes involved in the CB cycle in *Chlamydomonas*. Of particular focus was the distribution between pyrenoid and thylakoid subcellular compartments. The authors also performed mathematical modeling of the fluxes and estimated kinetic and transport rates between the compartments.

What is new about this work? My general assessment is that not much is new and what is new is not that surprising. The localization of the entire set of CBC enzymes is new, but most enzymes were found to be restricted to the stroma of the thylakoid. The results support what was previously known from other studies. Since Rubisco is known to be in both locations, it is obvious that the substrate and products of this enzyme will have to exchange between the compartments.

1) I don't think the kinetic modeling will give correct information because it is lacks important known regulation (i.e. allostery, light) and is not parameterized with enough data. For estimating more parameters (65 k values and 40 diffusion constants) from steady-state measurements of 11 total metabolite concentrations and 12 enzymes under 2 conditions, which by my calculations is 46 measurements.

2) The lack of statistical agreement for 6/16 metabolites, suggests the model (or the measurements) is wrong. Therefore, I don't understand why one would trust in the results from the model.

Once the experimental results of localization are known, I don't understand why the modeling would include transport reactions of all intermediates and enzymes in both locations. The modeling effort could be more focused if the subcellular experiments were first used as constraints, which would eliminate lots of transport and kinetic parameters.

The thermodynamic analysis while correct didn't provide much insight. What is the significance of two enzymes having a positive ∆G? In panel A, TPI has a positive ∆G. I think for a reversibility analysis in vivo metabolic flux analysis by 13C would be more informative.

The analysis of diffusion between compartments is important, but is rather unclear. In the subsection “Mechanisms of metabolite transport between microcompartments”, what is the estimated concentration gradient (it seems you give this in the fourth paragraph of the Discussion on the order of 24 or 8 nM)? Given a reasonable estimate for diffusivity of small molecules in water, is diffusion feasible to support the flux? I question the logic in the aforementioned paragraph. Just because concentrations go downhill doesn't rule out facilitated transport (i.e. not bulk diffusion), but the converse would be true (i.e. uphill requires expenditure of cellular energy). For example, one normally starts a microbial culture with 30g/L of glucose (160 mM), the intracellular concentration is certainly lower than this, but it doesn't mean diffusion is sufficient to explain the results! Certainly facilitated transport is needed to pass through the plasma membrane. The authors get the point of facilitated transport correct in the Discussion, but the results can be improved.

The conclusion that transport is tightly regulated is a flawed conclusion. Your modeling effort enforces a steady-state, so of course the export rate of 3PGA would have to balance the import of RuBP by a factor of 2. You are proving what you assume. Where is the evidence of regulation, particularly of transport?

---

## [Author Response]

Essential revisions:1) There are issues surrounding the potential localization and quantification of this partitioning for Rubisco that need to be discussed/tested depending on how the study was conducted.

We added a whole paragraph (subsection “Distribution of Calvin-Benson cycle enzymes”, last paragraph) to the Results section dealing with Rubisco localization. We also want to stress that our interpretation of the two models focusses on the differences of non-CCM and CCM-induced conditions and these differences in Rubisco localisation were found under all experimental set-ups and with both techniques (enzyme activity measurement of pyrenoid-enriched fraction and fluorescence analysis).

2) Better clarity on how fractional metabolite levels were estimated.

We did this by clarifying the Results (subsection “Metabolites in high-CO_2_ grown (HC) cells, and in low-CO_2_ grown cells after 15 min of exposure to 5% CO_2_ (LC*)), Discussion (first paragraph) and Materials and methods section (subsection "Metabolite measurements by HPLC-MS/MS” and subsection “Model parameterization”, last paragraph). We also improved the legend of Figure 2 and refer more often to the Supplementary file 1D (subsection “Metabolites in high-CO_2_ grown (HC) cells, and in low-CO_2_ grown cells after 15 min of exposure to 5% CO_2_ (LC*)”, first paragraph), where all the numbers are given and the reader can directly retrace our normalisation for example by recalculation.

3) There was concern that the lack of allosteric factors in the model was creating the lack of fitting on specific metabolites. This potential hindrance of the model should be discussed and the potential role of this in limiting the model fit should be commented upon.

Our detailed opinion on this issue is given as response to reviewer 3 (below). We also emphasized in the text that we used part of the experimental data to reduce the number of parameters of the model already and were therefore able to construct a very reliable, well-fitted model.

Reviewer #1:[…] The authors measure Rubisco activity in fractions enriched in pyrenoid and stromal proteins and find differences in the activity in both fractions that change in their ratio, depending on the CO_2_ level (Figure 2). The authors write that more than 60% of Rubisco is in the pyrenoid in LC cells (ambient CO_2_ level) and about 20% in HC cells (5% CO_2_ level). Then, they perform immunolocalization studies with Venus-fusion constructs (Figure 3). To my mind, the results shown in Figure 3 contradict the former findings of the authors. The small Rubisco subunits RBCS1 and RBCS2 are only visible in the pyrenoid and not in the other parts of the chloroplast, including the stroma. Unfortunately, the authors do not mention if the used cells were grown under LC or HC for these experiments. In any case, a certain portion of Rubisco should be distributed over the chloroplast stroma if Rubisco is present to a significant extent in the stroma, and the results are not an artifact of an impure stromal fraction. The stromal fraction was obtained by lysis of the cells and a centrifugation step using the resulting supernatant. I am afraid that this fraction may be contaminated with pyrenoid proteins that derive from partial lysis of the pyrenoid when disrupting the cells. Figure 1A in Mackinder et al. (2016), to which the authors refer, shows parts of pyrenoids as well, being in agreement with this possibility. An alternative would be that the fusion construct is not reflecting the real localization pattern of the smaIl Rubisco subunits. I suggest that the authors clarify this important step by conducting immunolocalization experiments with cells grown at LC and HC using anti-Rubisco antibodies before they draw their final conclusions. Especially under HC, it should be visible that the Rubisco is not only present in the pyrenoid but also in the rest of the plastid.

We understand the reviewer’s concern and added a whole paragraph (subsection “Distribution of Calvin-Benson cycle enzymes”, last paragraph) to the Results section to deal with the, at first sight, contracting results of enzyme activity measurements and the fluorescence analysis. We also added the growth conditions to the Results section (subsection “Distribution of Calvin-Benson cycle Enzymes”, third paragraph) and the legend of Figure 3, and added three paragraphs to the Materials and methods section (subsections “Enzymatic activities”, “Proteomics data” and “Protein Localization”) for more details.

We want to point out that the mentioned Figure 1A of Mackinder et al. (2016) indeed shows TEMs of pyrenoids and parts of pyrenoids. However, these sections were taken from the pellet fraction (the pyrenoid-enriched fraction) not the supernatant fraction.

Another critical point is that the authors change the CO_2_ treatment conditions when performing the metabolites studies. In the latter case, they use LC cells that have been treated for 15 min with 5% CO_2_ before harvesting. Their reasons are understandable, but it would be desirable to have the same conditions for the protein and metabolites studies.

Figure 2—figure supplement 1 shows that the physiology of the pyrenoid was not affected by the 15 min high CO_2_ exposure. Also, the transcriptome study of a shift from HC to LC by Brueggeman et al. (2012) found major changes of the transcriptome later than 30 min after the shift. To our knowledge, there is no indication that the shift from LC to HC is faster than this.

Brueggeman AJ, Gangadharaiah DS, Cserhati MF, Casero D, Weeks DP, Ladunga I (2012) Activation of the carbon concentrating mechanism by CO_2_ deprivation coincides with massive transcriptional restructuring in *Chlamydomonas reinhardtii*. Plant Cell 24: 1860-1875

In summary, this is an interesting and highly important topic and the manuscript should be of interest to a broad readership. However, the authors have to address the mentioned concerns to either verify their conclusion or to change it and consequently adapt their model.

Thanks a lot, we clarified our conclusion and methods.

Reviewer #2:[…] In my opinion the authors should address two issues near the beginning of the Discussion. The most serious issue relates to the estimation of metabolite concentrations. If I understand the methods correctly, the authors have quantitated the metabolite amount in the entire cell and have not tried to determine the metabolite amounts in the chloroplast versus cytoplasm. If that is true, it is not clearly stated in the beginning of the Discussion. How were chloroplast CBC metabolite levels estimated if the measured amount of each metabolite was for the entire cell? The authors need to say exactly what assumptions they used to calculate the amount of the metabolite in the stroma. The only place this was addressed was in the legend to Figure 2. It should be in the Materials and methods section and reiterated at the beginning of the discussion.

We added more information of the normalization of the metabolite concentration in the Results section (subsection “Metabolites in high-CO_2_ grown (HC) cells, and in low-CO_2_ grown cells after 15 min of exposure to 5% CO_2_ (LC*)) and clarified the beginning of the Discussion (first paragraph). Also, more details are given in the legend of Figure 2 and in the Materials and methods section (subsection "Metabolite measurements by HPLC-MS/MS”).

Additionally, the concentration of metabolites estimated from the modelling were compared to the concentration of metabolites in the chloroplast calculated based on whole cell measurements. In order to compare subcompartment-specific concentrations predicted by the model and calculated chloroplast concentrations, we used the sum of predicted stroma and pyrenoid concentration. We added a sentence to the Materials and methods section (subsection “Model parameterization”, last paragraph) to make this clear as well.

My second concern was which value for the percentage of Rubisco in the pyrenoid was used? Clearly the 60% pyrenoid localization seen in fractionation is lower than earlier estimates using immunogold or the fluorescence measurements. Again at the beginning of the model description and in the discussion the actual values used should be stressed. From Figure 1 it would appear that the value used was 100% in the pyrenoid but I am not certain. I think any value over 90% is reasonable but the authors should state why they favored the fluorescence method or literature values over the MS or activity measurements. I suspect there is some dissolution of the pyrenoid and that is what accounts for the discrepancy but it should be noted.

For model parametrization we wanted to use values coming from the same study and therefore, an average of 61.8% pyrenoid localization of Rubisco under LC as measured in this study was used. The added paragraph (subsection “Distribution of Calvin-Benson cycle enzymes”) discusses the possibility of dissolution.

The actual values were added to the text in the Results (subsection “Distribution of Calvin-Benson cycle enzymes”, last paragraph and subsection “Modelling the”, fifth paragraph) and the Discussion (first paragraph) section.

Reviewer #3:[…] What is new about this work? My general assessment is that not much is new and what is new is not that surprising. The localization of the entire set of CBC enzymes is new, but most enzymes were found to be restricted to the stroma of the thylakoid. The results support what was previously known from other studies. Since Rubisco is known to be in both locations, it is obvious that the substrate and products of this enzyme will have to exchange between the compartments.1) I don't think the kinetic modeling will give correct information because it is lacks important known regulation (i.e. allostery, light) and is not parameterized with enough data.

We agree with the reviewer that mass action may not be the most realistic enzyme kinetics, however, in absence of information about the kinetic form of the flux (e.g. Michaelis-Menten, power law, or other special form) and to allow for a systematic investigation, we aimed to focus on the most mathematically tractable approach, hence the usage of mass action kinetic.

In addition, every reaction is split into its elementary reactions, which includes the direct effect of enzymes. Therefore, the approach can account for variable reaction rates across different conditions due to changes in enzyme concentrations.

Furthermore, we would like to point out that there are examples of well-studied highly regulated metabolic processes (e.g. the Calvin-Benson cycle) which can be accurately modelled by mass action kinetic (with apparent rate constants, see comprehensive model comparison of Arnold and Nikoloski (2011, Trends in Plant Science).

For estimating more parameters (65 k values and 40 diffusion constants) from steady-state measurements of 11 total metabolite concentrations and 12 enzymes under 2 conditions, which by my calculations is 46 measurements.

To reduce the number parameters estimated we used the information about enzyme localization. Based on the experimental data we reduced the number of diffusion constants from 40 to 2. In addition, the number of reactions in pyrenoid is adapted,for example in the scenario were Rubisco is the only enzyme in the pyrenoid, diffusion is allowed for RuBP and PGA only. We added a sentence to the subsection “Modelling the effect of chloroplast microcompartmentation on the CBC” to make this clear.

2) The lack of statistical agreement for 6/16 metabolites, suggests the model (or the measurements) is wrong. Therefore, I don't understand why one would trust in the results from the model.

We do not agree with the reviewer’s comment that the model is wrong. Model fit is usually validated based on the overall chi square value, which provided significant fit for the samples analysed. Therefore, we are convinced that our model is suited to be used for hypothesis testing.

Individual chi square values were calculated since we wanted to know which metabolites fit better or worse and if these metabolites have a particular role in the modelled processes.

Indeed, we found the metabolites involved in reactions for which it is known that they do not follow mass action kinetics not to fit well. Nevertheless, we would like to stress that the overall model fits were statistically significant.

We hope the reviewer agrees that for the question of interest an overall fit between model predictions and measurements is sufficient.

Once the experimental results of localization are known, I don't understand why the modeling would include transport reactions of all intermediates and enzymes in both locations. The modeling effort could be more focused if the subcellular experiments were first used as constraints, which would eliminate lots of transport and kinetic parameters.

As suggested by the reviewer, we indeed used the experimental data to reduce the number of model reactions and updated the text to make this clear.

The thermodynamic analysis while correct didn't provide much insight. What is the significance of 2 enzymes having a positive ∆G? In panel A, TPI has a positive ∆G. I think for a reversibility analysis in vivo metabolic flux analysis by 13C would be more informative.

We thank the reviewer for recognizing this point. We updated the text accordingly (subsection “Two modes of CBC operation”).

The analysis of diffusion between compartments is important, but is rather unclear. In the subsection “Mechanisms of metabolite transport between microcompartments”, what is the estimated concentration gradient (it seems you give this in the fourth paragraph of the Discussion on the order of 24 or 8 nM)? Given a reasonable estimate for diffusivity of small molecules in water, is diffusion feasible to support the flux? I question the logic in the aforementioned paragraph.

We added the values to make the difference directly visible (subsection “Mechanisms of metabolite transport between microcompartments”).

Just because concentrations go downhill doesn't rule out facilitated transport (i.e. not bulk diffusion), but the converse would be true (i.e. uphill requires expenditure of cellular energy). For example, one normally starts a microbial culture with 30g/L of glucose (160 mM), the intracellular concentration is certainly lower than this, but it doesn't mean diffusion is sufficient to explain the results! Certainly facilitated transport is needed to pass through the plasma membrane. The authors get the point of facilitated transport correct in the Discussion, but the results can be improved.

We agree with the reviewer that a downhill concentration gradient does not directly mean diffusion and therefore, we updated the text accordingly.

We improved the usage of diffusion in the Results section (subsection “Mechanisms of metabolite transport between microcompartments”).

The conclusion that transport is tightly regulated is a flawed conclusion. Your modeling effort enforces a steady-state, so of course the export rate of 3PGA would have to balance the import of RuBP by a factor of 2. You are proving what you assume. Where is the evidence of regulation, particularly of transport.

The reviewer is correct, this observation is trivial and therefore, we opted to remove this sentence.